# A Self-Tuning Actor-Critic Algorithm

**Tom Zahavy, Zhongwen Xu, Vivek Veeriah, Matteo Hessel, Junhyuk Oh,**
**Hado van Hasselt, David Silver and Satinder Singh**
**Deepmind**
{tomzahavy,zhongwen,vveeriah,mtthss,junhyuk,hado,davidsilver,baveja}@google.com

## Abstract

Reinforcement learning algorithms are highly sensitive to the choice of hyperparameters, typically requiring significant manual effort to identify hyperparameters that perform well on a new domain. In this paper, we take a step towards addressing this issue by using metagradients to automatically adapt hyperparameters online by meta-gradient descent (Xu et al., 2018). We apply our algorithm, Self-Tuning Actor-Critic (STAC), to self-tune all the differentiable hyperparameters of an actor-critic loss function, to discover auxiliary tasks, and to improve off-policy learning using a novel leaky V-trace operator. STAC is simple to use, sample efficient and does not require a significant increase in compute. Ablative studies show that the overall performance of STAC improved as we adapt more hyperparameters. When applied to the Arcade Learning Environment (Bellemare et al. 2012), STAC improved the median human normalized score in 200M steps from $243\%$ to $364\%$. When applied to the DM Control suite (Tassa et al., 2018), STAC improved the mean score in 30M steps from 217 to 389 when learning with features, from 108 to 202 when learning from pixels, and from 195 to 295 in the Real-World Reinforcement Learning Challenge (Dulac-Arnold et al., 2020).

## 1 Introduction

Deep Reinforcement Learning (RL) algorithms often have many modules and loss functions with many hyperparameters. When applied to a new domain, these hyperparameters are searched via cross-validation, random search (Bergstra & Bengio, 2012), or population-based training (Jaderberg et al., 2017), which requires extensive computing resources. Meta-learning approaches in RL (e.g., MAML, Finn et al. (2017)) focus on learning good initialization via multi-task learning and transfer. However, many of the hyperparameters must be adapted during the agent's lifetime to achieve good performance (learning rate scheduling, exploration annealing, etc.). This motivates a significant body of work on specific solutions to tune specific hyperparameters, within a single agent *lifetime* (Schaul et al., 2019; Mann et al., 2016; White & White, 2016; Rowland et al., 2019; Sutton, 1992).

Metagradients, on the other hand, provide a general and compute-efficient approach for self-tuning in a single lifetime. The general concept is to represent the training loss as a function of both the agent parameters and the hyperparameters. The agent optimizes the parameters to minimize this loss function, w.r.t the current hyperparameters. The hyperparameters are then self-tuned via backpropagation to minimize a fixed loss function. This approach has been used to learn the discount factor or the $\lambda$ coefficient (Xu et al., 2018), to discover intrinsic rewards (Zheng et al., 2018) and auxiliary tasks (Veeriah et al., 2019). Finally, we note that there also exist derivative-free approaches for self-tuning hyper parameters (Paul et al., 2019; Tang & Choromanski, 2020).

This paper makes the following contributions. **First,** we introduce two novel ideas that extend IMPALA (Espeholt et al., 2018) with additional components. **(1)** The first agent, referred to as a Self-Tuning Actor-Critic (STAC), self-tunes all the differentiable hyperparameters in the IMPALA loss function. In addition, STAC introduces a *leaky V-trace* operator that mixes importance sampling

(IS) weights with truncated IS weights. The mixing coefficient in leaky V-trace is differentiable (unlike the original V-trace) but similarly balances the variance-contraction trade-off in off-policy learning. **(2)** The second agent, *STACX* (STAC with auXiliary tasks), adds auxiliary parametric actor-critic loss functions to the loss function and self-tunes their metaparameters. STACX self-tunes the discount factors of these auxiliary losses to different values than those of the main task, helping it to reason about multiple horizons.

**Second,** we demonstrate empirically that self-tuning consistently improves performance. When applied to the Arcade Learning Environment (Bellemare et al., 2013, ALE), STAC improved the median human normalized score in 200M steps from $243\%$ to $364\%$. When applied to the DM Control suite (Tassa et al., 2018), STAC improved the mean score in 30M steps from 217 to 389 when learning with features, from 108 to 202 when learning from pixels, and from 195 to 295 in the Real-World Reinforcement Learning Challenge (Dulac-Arnold et al., 2020).

We conduct extensive ablation studies, showing that the performance of STACX consistently improves as it self-tunes more hyperparameters; and that STACX improves the baseline when self-tuning different subsets of the metaparameters. STACX performs considerably better than previous metagradient algorithms (Xu et al., 2018; Veeriah et al., 2019) and across a broader range of environments.

**Finally,** we investigate the properties of STACX via a set of experiments. (1) We show that STACX is more robust to its hyperparameters than the IMPALA baseline. (2) We visualize the self-tuned metaparameters through training and identify trends. (3) We demonstrate a tenfold scale up in the number of self-tuned hyperparameters – 21 compared to two in (Xu et al., 2018). This is the most significant number of hyperparameters tuned by meta-learning at scale and does not require a significant increase in compute (see Table 4 in the supplementary and the discussion that follows it).

## 2 Background

We begin with a brief introduction to actor-critic algorithms and IMPALA (Espeholt et al., 2018). Actor-critic agents maintain a policy $\pi_\theta(a|x)$ and a value function $V_\theta(x)$ that are parameterized with parameters $\theta$. These policy and the value function are trained via an actor-critic update rule, with a policy gradient loss and a value prediction loss. In IMPALA, we additionally add an entropy regularization loss. The update is represented as the gradient of the following pseudo-loss function

$$
\begin{aligned}
L_{\text{Value}}(\theta) &= \sum_{s \in \text{T}} \left( v_s - V_\theta(x_s) \right)^2 \\
L_{\text{Policy}}(\theta) &= -\sum_{s \in \text{T}} \rho_s \log \pi_\theta(a_s|x_s)(r_s + \gamma v_{s+1} - V_\theta(x_s)) \\
L_{\text{Entropy}}(\theta) &= -\sum_{s \in \text{T}} \sum_a \pi_\theta(a|x_s) \log \pi_\theta(a|x_s) \\
L(\theta) &= g_v L_{\text{Value}}(\theta) + g_p L_{\text{Policy}}(\theta) + g_e L_{\text{Entropy}}(\theta).
\end{aligned} \tag{1}
$$

In each iteration $t$, the gradients of these losses are computed on data T that is composed from a mini batch of $m$ trajectories, each of size $n$ (see the the supplementary material for more details). We refer to the policy that generates this data as the behaviour policy $\mu(a_s|x_s)$, where the superscript $s$ will refer to the time index within a trajectory. In the on policy case, $\mu(a_s|x_s) = \pi(a_s|x_s)$, $\rho_s = 1$, and we have that $v_s$ is the n-steps bootstrapped return $v_s = \sum_{j=s}^{s+n-1} \gamma^{j-s} r_j + \gamma^n V(x_{s+n})$.

IMPALA uses a distributed actor critic architecture, that assigns copies of the policy parameters to multiple actors in different machines to achieve higher sample throughput. As a result, the target policy $\pi$ on the learner machine can be several updates ahead of the actor's policy $\mu$ that generated the data used in an update. Such off policy discrepancy can lead to biased updates, requiring us to multiply the updates with importance sampling (IS) weights for stable learning. Specifically, IMPALA (Espeholt et al., 2018) uses truncated IS weights to balance the variance-contraction trade-off on these off-policy updates. This corresponds to instantiating Eq. (1) with

$$
\begin{aligned}
v_s &= V(x_s) + \sum_{j=s}^{s+n-1} \gamma^{j-s} \left( \Pi_{i=s}^{j-1} c_i \right) \delta_j V, \quad \delta_j V = \rho_j (r_j + \gamma V(x_{j+1}) - V(x_j)) \\
\rho_j &= \min \left( \bar{\rho}, \frac{\pi(a_j|x_j)}{\mu(a_j|x_j)} \right), c_i = \lambda \min \left( \bar{c}, \frac{\pi(a_i|x_i)}{\mu(a_i|x_i)} \right).
\end{aligned} \tag{2}
$$

**Metagradients.** In the following, we consider three types of parameters: $\theta$ – the agent parameters; $\zeta$ – the hyperparameters; $\eta \subset \zeta$ – the metaparameters. $\theta$ denotes the parameters of the **agent** and parameterizes, for example, the value function and the policy; these parameters are randomly initialized at the beginning of an agent's lifetime and updated using backpropagation on a suitable *inner* loss function. $\zeta$ denotes the **hyperparameters**, including, for example, the parameters of the optimizer (e.g., the learning rate) or the parameters of the loss function (e.g., the discount factor); these may be tuned throughout many lifetimes (for instance, via random search) to optimize an *outer* (validation) loss function. Typical deep RL algorithms consider only these first two types of parameters. In metagradient algorithms a third set of parameters is specified: the **metaparameters**, denoted $\eta$, which are a *subset* of the differentiable parameters in $\zeta$. Starting from some initial value (itself a hyperparameter), they are then self-tuned during training within a single lifetime.

Metagradients are a general framework for adapting, online, within a single lifetime, the differentiable hyperparameters $\eta$. Consider an inner loss that is a function of both the parameters $\theta$ and the metaparameters $\eta$: $L_{\text{inner}}(\theta; \eta)$. On each step of an inner loop, $\theta$ can be optimized with a fixed $\eta$ to minimize the inner loss $L_{\text{inner}}(\theta; \eta)$, by updating $\theta$ with the following gradient $\tilde{\theta}(\eta_t) \doteq \theta_{t+1} = \theta_t - \nabla_\theta L_{\text{inner}}(\theta_t; \eta_t)$.

In an outer loop, $\eta$ can then be optimized to minimize the outer loss by taking a metagradient step. As $\tilde{\theta}(\eta)$ is a function of $\eta$ this corresponds to updating the $\eta$ parameters by differentiating the outer loss w.r.t $\eta$, such that $\eta_{t+1} = \eta_t - \nabla_\eta L_{\text{outer}}(\tilde{\theta}(\eta_t))$. The algorithm is general and can be applied, in principle, to any differentiable meta-parameter $\eta$ used by the inner loss. Explicit instantiations of the metagradient RL framework require the specification of the inner and outer loss functions.

## 3  Self-Tuning actor-critic agents

We now describe the inner and outer loss of our agent. The general idea is to self-tune all the differentiable hyperparameters. The outer loss is the original IMPALA loss (Eq. (1)) with an additional Kullback–Leibler (KL) term, which regularizes the $\eta$-update not to change the policy:

$$L_{\text{outer}}(\tilde{\theta}(\eta)) = g_v^{\text{outer}} L_{\text{Value}}(\theta) + g_p^{\text{outer}} L_{\text{Policy}}(\theta) + g_e^{\text{outer}} L_{\text{Entropy}}(\theta) + g_{\text{kl}}^{\text{outer}} \text{KL}(\pi_{\tilde{\theta}(\eta)}, \pi_\theta). \quad (3)$$

The inner loss function, is parametrized by the metaparameters $\eta = \{\gamma, \lambda, g_v, g_p, g_e\}$:

$$L(\theta; \eta) = g_v L_{\text{Value}}(\theta) + g_p L_{\text{Policy}}(\theta) + g_e L_{\text{Entropy}}(\theta), \quad (4)$$

Notice that $\gamma$ and $\lambda$ affect the inner loss through the definition of $v_s$ in Eq. (2). The loss coefficients $g_v, g_p, g_e$ allow for loss specific learning rates and support dynamically balancing exploration with exploitation by adapting the entropy loss weight. We apply a sigmoid activation on all the metaparameters, which ensures that they remain bounded. We also multiply the loss coefficients $(g_v, g_e, g_p)$ by the respective coefficient in the outer loss to guarantee that they are initialized from the same values. For example, $\gamma = \sigma(\gamma), g_v = \sigma(g_v) g_v^{\text{outer}}$. The exact details can be found in the supplementary (Algorithm 2, line 11).

Table 1 summarizes all the **hyperparameters** that are required for STAC. STAC has new hyperparameters (compared to IMPALA), but we found that using simple "rules of thumb" is sufficient to tune them. These include the initializations of the metaparameters, the hyperparameters of the outer loss, and meta optimizer parameters. The exact values can be found in the supplementary (see Table 3. For the outer loss hyperparameters, we use exactly the same hy-

| | IMPALA | STAC |
|---|---|---|
| $\theta$ | $V_\theta, \pi_\theta$ | $V_\theta, \pi_\theta$ |
| $\zeta$ | $\{\gamma, \lambda, g_v, g_p, g_e\}$ | $\{\gamma^{\text{outer}}, \lambda^{\text{outer}}, g_v^{\text{outer}}, g_p^{\text{outer}}, g_e^{\text{outer}}\}$ Initialisations Meta optimizer parameters, $g_{\text{kl}}^{\text{outer}}$ |
| $\eta$ | – | $\{\gamma, \lambda, g_v, g_p, g_e\}$ |

Table 1: Parameters in IMPALA and STAC.

perparameters that were used in the IMPALA paper for all of our agents ($g_v^{\text{outer}} = 0.25, g_p^{\text{outer}} = 1, g_v^{\text{outer}} = 1, \lambda^{\text{outer}} = 1$), with one exception: we use $\gamma = 0.995$ as it was found in (Xu et al., 2018) to improve in Atari the performance of IMPALA and the metagradient agent in Atari, and $\gamma = 0.99$ in DM control suite.

For the initializations of the metaparameters we use the corresponding parameters in the outer loss, i.e., for any metaparameter $\eta_i$, we set $\eta_i^{\text{Init}} = 4.6$ such that $\sigma(\eta_i^{\text{Init}}) = 0.99$. This guarantees that the inner loss is initialized to be (almost) the same as the outer loss. The exact value was chosen arbitrarily, and we later show in Fig. 4(c) that the algorithm is not sensitive to it. For the meta optimizer, we use ADAM with default settings (e.g., learning rate is set to $10^{-3}$), and for the the KL coefficient, we use $g_{\text{kl}}^{\text{outer}} = 1$).

## 3.1 STAC and leaky V-trace

The hyperparameters that we considered for self-tuning so far, $\eta = \{\gamma, \lambda, g_v, g_p, g_e\}$, parametrized the loss function in a differentiable manner. The truncation levels in the V-trace operator, on the other hand, are nondifferentiable. We now introduce the Self-Tuning Actor-Critic (STAC) agent. STAC self-tunes a variant of the V-trace operator that we call leaky V-trace (in addition to the previous five meta parameters), motivated by the study of nonlinear activations in Deep Learning (Xu et al., 2015). Leaky V-trace uses a leaky rectifier (Maas et al., 2013) to truncate the importance sampling weights, where a differentiable parameter controls the leakiness. Moreover, it provides smoother gradients and prevents the unit from getting saturated.

Before we introduce Leaky V-trace, let us first recall how the off-policy trade-offs are represented in V-trace using the coefficients $\bar{\rho}, \bar{c}$. The weight $\rho_t = \min(\bar{\rho}, \frac{\pi(a_t|x_t)}{\mu(a_t|x_t)})$ appears in the definition of the temporal difference $\delta_t V$ and defines the fixed point of this update rule. The fixed point of this update is the value function $V^{\pi_{\bar{\rho}}}$ of the policy $\pi_{\bar{\rho}}$ that is somewhere between the behavior policy $\mu$ and the target policy $\pi$ controlled by the hyperparameter $\bar{\rho}$,

$$\pi_{\bar{\rho}} = \frac{\min\left(\bar{\rho}\mu(a|x), \pi(a|x)\right)}{\sum_b \min\left(\bar{\rho}\mu(b|x), \pi(b|x)\right)}.$$

The product of the weights $c_s, ..., c_{t-1}$ in Eq. (2) measures how much a temporal difference $\delta_t V$ observed at time $t$ impacts the update of the value function. The truncation level $\bar{c}$ is used to control the speed of convergence by trading off the update variance for a larger contraction rate. The variance associated with the update rule is reduced relative to importance-weighted returns by clipping the importance weights. On the other hand, the clipping of the importance weights effectively cuts the traces in the update, resulting in the update placing less weight on later TD errors and worsening the contraction rate of the corresponding operator. Following this interpretation of the off policy coefficients, we propose a variation of V-trace which we call *leaky V-trace* with parameters $\alpha_\rho \geq \alpha_c$,

$$\text{IS}_t = \frac{\pi(a_t|x_t)}{\mu(a_t|x_t)}, \quad \rho_t = \alpha_\rho \min\left(\bar{\rho}, \text{IS}_t\right) + (1 - \alpha_\rho)\text{IS}_t, \quad c_i = \lambda\left(\alpha_c \min\left(\bar{c}, \text{IS}_t\right) + (1 - \alpha_c)\text{IS}_t\right),$$

$$v_s = V(x_s) + \sum_{t=s}^{s+n-1} \gamma^{t-s}\left(\Pi_{i=s}^{t-1}c_i\right)\delta_t V, \quad \delta_t V = \rho_t(r_t + \gamma V(x_{t+1}) - V(x_t)). \tag{5}$$

We highlight that for $\alpha_\rho = 1, \alpha_c = 1$, Leaky V-trace is exactly equivalent to V-trace, while for $\alpha_\rho = 0, \alpha_c = 0$, it is equivalent to canonical importance sampling. For other values, we get a mixture of the truncated and not-truncated importance sampling weights.

Theorem 1 below suggests that Leaky V-trace is a contraction mapping and that the value function that it will converge to is given by $V^{\pi_{\bar{\rho}, \alpha_\rho}}$, where

$$\pi_{\bar{\rho}, \alpha_\rho} = \frac{\alpha_\rho \min\left(\bar{\rho}\mu(a|x), \pi(a|x)\right) + (1 - \alpha_\rho)\pi(a|x)}{\alpha_\rho \sum_b \min\left(\bar{\rho}\mu(b|x), \pi(b|x)\right) + 1 - \alpha_\rho},$$

is a policy that mixes (and then re-normalizes) the target policy with the V-trace policy. We provide a formal statement of Theorem 1, and detailed proof in the supplementary material (Section 10).

**Theorem 1.** *The leaky V-trace operator defined by Eq.* (5) *is a contraction operator, and it converges to the value function of the policy defined above.*

Similar to $\bar{\rho}$, the new parameter $\alpha_\rho$ controls the fixed point of the update rule and defines a value function that interpolates between the value function of the target policy $\pi$ and the behavior policy $\mu$. Specifically, the parameter $\alpha_c$ allows the importance weights to "leak back" creating the opposite effect to clipping. Since Theorem 1 requires us to have $\alpha_\rho \geq \alpha_c$, our main STAC implementation

parametrises the loss with a single parameter $\alpha = \alpha_\rho = \alpha_c$. In addition, we also experimented with a version of STAC that learns both $\alpha_\rho$ and $\alpha_c$. This variation of STAC learns the rule $\alpha_\rho \geq \alpha_c$ on its own (see Fig. 5(b)). Note that low values of $\alpha_c$ lead to importance sampling, which is high contraction but high variance. On the other hand, high values of $\alpha_c$ lead to V-trace, which is lower contraction and lower variance than importance sampling. Thus exposing $\alpha_c$ to meta-learning enables STAC to control the contraction/variance trade-off directly.

In summary, the **metaparameters** for STAC are $\{\gamma, \lambda, g_v, g_p, g_e, \alpha\}$. To keep things simple, when using Leaky V-trace we make two simplifications w.r.t the **hyperparameters**. First, we use V-trace to initialise Leaky V-trace, i.e., we initialise $\alpha = 1$. Second, we fix the outer loss to be V-trace, i.e. we set $\alpha^{\text{outer}} = 1$.

### 3.2 STAC with auxiliary tasks (STACX)

Next, we introduce a new agent, that extends STAC with auxiliary policies, value functions, and corresponding auxiliary loss functions. Auxiliary tasks have proven to be an effective solution to learning useful representations from limited amounts of data. We observe that each set of meta-parameters induces a separate inner loss function, which can be thought of as an auxiliary loss. To meta-learn auxiliary tasks, STACX self-tunes additional sets of meta-parameters, independently of the main head, but via the same meta-gradient mechanism. The novelty here comes from STACX's ability to discover the auxiliary tasks most useful to it. E.g., the discount factors of these auxiliary losses allow STACX to reason about multiple horizons.

STACX's architecture has a shared representation layer $\theta_{\text{shared}}$, from which it splits into $n$ different heads (Section 3.2). For the shared representation layer we use the deep residual net from (Espeholt et al., 2018). Each head has a policy and a corresponding value function that are represented using a 2 layered MLP with parameters $\{\theta_i\}_{i=1}^n$. Each one of these heads is trained in the inner loop to minimize a loss function $L(\theta_i; \eta_i)$, parametrized by its own set of metaparameters $\eta_i$.

The STACX agent policy is defined as the policy of a specific head ($i = 1$). We considered two more variations that allow the other heads to act. Both did not work well, and we provide more details in the supplementary (Section 9.3). The hyperparameters $\{\eta_i\}_{i=1}^n$ are trained in the outer loop to improve the performance of this single head. Thus, the role of the auxiliary heads is to act as auxiliary tasks (Jaderberg et al., 2016) and improve the shared representation $\theta_{\text{shared}}$. Finally, notice that each head has its own policy $\pi^i$, but the behavior policy is fixed to be $\pi^1$. Thus, to optimize the auxiliary heads, we use (Leaky) V-trace for off-policy corrections.

The **metaparameters** for STACX are $\{\gamma^i, \lambda^i, g_v^i, g_p^i, g_e^i, \alpha^i\}_{i=1}^3$. Since the outer loss is defined only w.r.t head 1, introducing the auxiliary tasks into STACX does not require new hyperparameters for the outer loss. In addition, we use the same initialization values for all the auxiliary tasks. Thus, STACX has the same **hyperparameters** as STAC.

**Summary.** In this Section, we showed how embracing self-tuning via metagradients enables us to introduce novel ideas into our agent. We augmented our agent with a parameterized Leaky V-trace operator and with *self-tuned* auxiliary loss functions. We did not have to tune these new hyperparameters because we relied on metagradients to self-tune them. We emphasize here that STACX is not a fully parameter-free algorithm. Nevertheless, we argue that STACX requires the same hyperparameter tuning as IMPALA, since we use default values for the new hyperparameters. We further evaluated these design principles in Fig. 4.

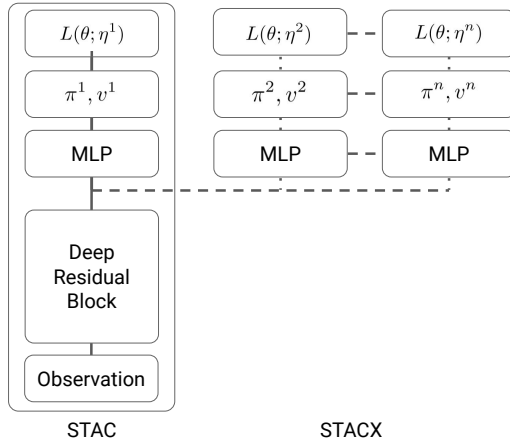

Figure 1: Block diagrams of STAC and STACX.

# 4 Experiments

## 4.1 Atari Experiments.

We begin the empirical evaluation of our algorithm in the Arcade Learning Environment (Bellemare et al., 2013, ALE). To be consistent with prior work, we use the same ALE setup that was used in (Espeholt et al., 2018; Xu et al., 2018); in particular, the frames are down-scaled and grayscaled.

Fig. 2(a) presents the normalized median scores during training, computed in the following manner: for each Atari game, we compute the human normalized score after 200M frames of training and average this over three seeds. We then report the overall median score over the 57 Atari domains for four variations of our algorithm STACX (blue, solid), STAC (green, solid), IMPALA with fixed auxiliary tasks (blue, dashed), and IMPALA (green, dashed). Inspecting Fig. 2(a) we observe two trends: using self-tuning improves the performance with/out auxiliary tasks (solid vs. dashed lines), and using auxiliary tasks improves the performance with/out self-tuning (blue vs. green lines). In the supplementary (Section 12), we report the relative improvement over IMPALA in individual games.

STACX outperforms all other agents in this experiment, achieving a median score of $364\%$, a new state of the art result in the ALE benchmark for training online model-free agents for 200M frames. In fact, there are only two agents that reported better performance after 200M frames: LASER (Schmitt et al., 2019) achieved a normalized median score of $431\%$ and MuZero (Schrittwieser et al., 2019) achieved $731\%$. These papers propose algorithmic modifications that are orthogonal to our approach and can be combined in future work; LASER combines IMPALA with a uniform large-scale experience replay; MuZero uses replay and a tree-based search with a learned model.

In Fig. 2(b), we perform an ablative study of our approach by training different variations of STAC (green) and STACX (blue). For each bar, we report the subset of metaparameters that are being self-tuned in this ablative study. The bottom bar for each color with $\{\}$ corresponds to not using self-tuning at all (IMPALA w/o auxiliary tasks), and the topmost color corresponds to self-tuning all the metaparameters (as reported in Fig. 2(a)). In between, we report results for tuning only subsets of the metaparameters. For example, $\eta = \{\gamma\}$ corresponds to self-tuning a single loss function where only $\gamma$ is self-tuned. When we do not self-tune a hyperparameter, its value is fixed to its corresponding value in the outer loss. For example, in all the ablative studies besides the two topmost bars, we do not self-tune $\alpha$, which means that we use V-trace instead (fix $\alpha = 1$). Finally, in red, we report results from different baselines as a point of reference (in this case, IMPALA is using $\gamma = 0.99$), and our variation of IMPALA (green, bottom) with $\gamma = 0.995$ indeed achieves higher score as was reported in (Xu et al., 2018). We also note that the metagradient agent of Xu et al. (2018) achieved higher performance than our variation of STAC that is only learning $\eta = \{\gamma, \lambda\}$. We further discuss this in the supplementary (Section 9.5).

Inspecting Fig. 2(b) we observe that the performance of STAC and STACX consistently improves as they self-tune more metaparameters. These metaparameters control different trade-offs in reinforcement learning: discount factor controls the effective horizon, loss coefficients affect learning rates, the Leaky V-trace coefficient controls the variance-contraction-bias trade-off in off-policy RL.

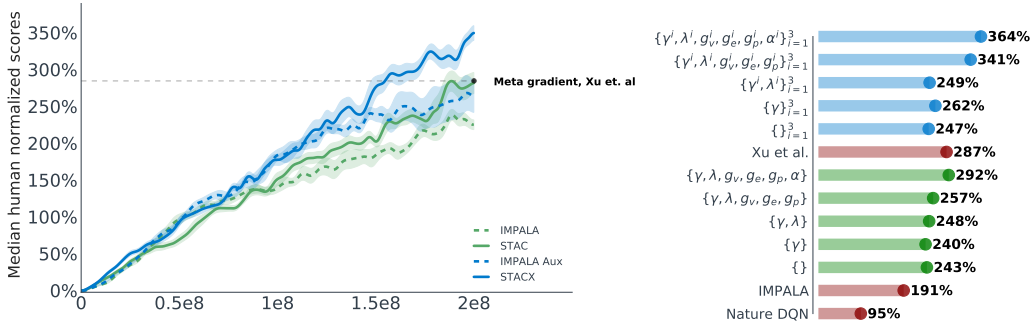

(a) Average learning curves with $0.5\cdot$ std confidence intervals.

(b) Ablative studies of STAC (green) and STACX (blue) alongside baselines (red).

Figure 2: Median normalized scores in 57 Atari games. Average over three seeds, 200M frames.

## 4.2 DM control suite

To further examine the generality of STACX we conducted a set of experiments in the DM control suite (Tassa et al., 2018). We considered three setups: (a) learning from feature observations, (b) learning from pixel observations, and (c) the real-world RL challenge (Dulac-Arnold et al., 2020, RWRL). The latter introduces a set of challenges (inspired by real-world scenarios) on top of existing control domains: delayed actions, observations and rewards, action repetition, added noise to the actions, stuck/dropped sensors, perturbations, and increased state dimensions. These challenges are combined in 3 difficulty levels (easy, medium, and hard) for humanoid, walker, quadruped, and cartpole. Scores are normalized to $[0, 1000]$ by the environment (Tassa et al., 2018).

We use the same algorithm and similar hyperparameters to the ones we use in the Atari experiments. For most of the hyperparameters (and in particular, those that are relevant to STACX) we use the same values as we used in the Atari experiments (e.g., $g_v, g_p, g_e$); others, like learning rate and discount factor, were re-tuned for the control domains (but remain fixed across all three setups). The exact details can be found in the supplementary (Section 9.3). For continuous actions, our network outputs two variables per action dimension that correspond to the mean and the standard deviation of a squashed Gaussian distribution (Haarnoja et al., 2018). The squashing refers to applying a tanh activation on the samples of the Gaussian, resulting in bounded actions. In addition, instead of using entropy regularization, we use a KL to standard Gaussian.

We emphasize here that while online actor-critic algorithms (IMPALA, A3C) do not achieve SOTA results in DM control, the results we present for IMPALA in Fig. 3(a) are consistent with the A3C results in (Tassa et al., 2018). The goal of these experiments is to measure the relative improvement from self-tuning. In Fig. 3, we average the results across suite domains and across three seeds. Standard deviation error bars w.r.t the seeds are reported in shaded areas. In the supplementary (Section 11) we provide domain-specific learning curves.

Inspecting Fig. 3 we observe two trends. **First**, using self-tuning improves performance (solid vs. dashed lines) in all three suites, w/o using the auxiliary tasks. **Second**, the auxiliary tasks improve performance when learning from pixels (Fig. 3(b)), which is consistent with the results in Atari. When learning from features (Fig. 3(a), Fig. 3(c)), we observe that IMPALA performs better without auxiliary tasks. This is reasonable, as there is less need for strong representation learning in this case. Nevertheless, STACX performs better than IMPALA as it can self-tune the loss coefficients of the auxiliary tasks to low values. Since this takes time, STACX performs worse than STAC.

Similar to the A3C baseline (using features), all of our agents were not able to solve the more challenging control domains (e.g., humanoid). Nevertheless, by using self-tuning, STAC, and STACX significantly outperformed the IMPALA baselines in many of the control domains. In the RWRL challenge, they even outperform strong baselines like D4PG and DMPO in the average score. Moreover, STAC was able to solve (to achieve an almost perfect score) two RWRL domains (quadruped.easy, cartpole.easy), making a new SOTA in these domains.

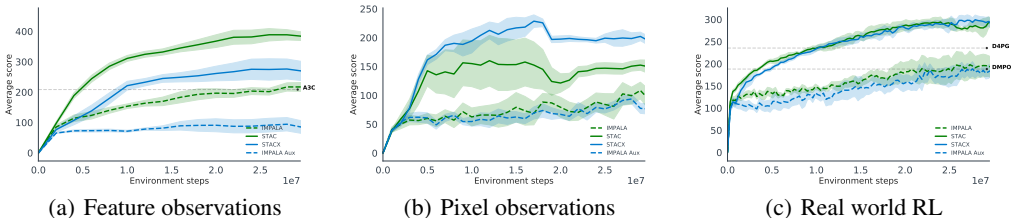

|  |  |  |
|:---:|:---:|:---:|
| (a) Feature observations | (b) Pixel observations | (c) Real world RL |

Figure 3: Aggregated results in DM Control of STACX, STAC and IMPALA with/out Auxiliary tasks. In dashed lines, we report the aggregated results of baselines at the end of training; A3C as reported in (Tassa et al., 2018), DMPO and D4PG as reported in (Dulac-Arnold et al., 2020).

## 4.3 Analysis

To better understand the behavior of the proposed method, we chose one domain, Atari, and performed some additional experiments. We begin by investigating the **robustness** of STACX to its

hyperparameters. First, we consider the hyperparameters of the **outer loss**, and compare the robustness of STACX with that of IMPALA. For each hyperparameter $(\gamma, g_v)$ we select 5 perturbations. For STACX we perturb the hyperparameter in the outer loss $(\gamma^{\text{outer}}, g_v^{\text{outer}})$ and for IMPALA we perturb the corresponding hyperparameter $(\gamma, g_v)$. We randomly selected 5 Atari games and presented the mean and standard deviation across 3 random seeds after 200M frames.

Fig. 4(a) and Fig. 4(b) present the results for the discount factor $\gamma$ and for $g_v$ respectively. We can see that overall, STACX performs better than IMPALA (in $72\%$ and $80\%$ of the setups, respectively). This is perhaps not surprising because we have already seen that STACX outperforms IMPALA in Atari, but now we observe this over a wider range of hyperparameters. In addition, we can see that in specific games, there are specific hyperparameter values that result in lower performance. In particular, in James Bond and Chopper Command (the two topmost rows), we observe lower performance when decreasing the discount factor $\gamma$ and when decreasing $g_v$. While the performance of both STACX and IMPALA deteriorates in these experiments, the effect on STACX is less severe.

In Fig. 4(c) we investigate the robustness of STACX to the **initialization** of the metaparameters. Since IMPALA does not have this hyperparameter, we only investigate its effect on STACX. We selected five different initialization values (all close to 1,, so the inner loss is close to the outer loss) and fixed all the other hyperparameters (e.g., the outer loss). Inspecting Fig. 4(c), we can see that the performance of STACX does not change too much when we change the value of the initializations, both in the case where we perturb the initializations of all the meta parameters (top), and only the discount (bottom). These observations confirm that our design choice to arbitrary initializing all the meta parameters to $0.99$ is sufficient, and there is no need to tune this new hyperparameter.

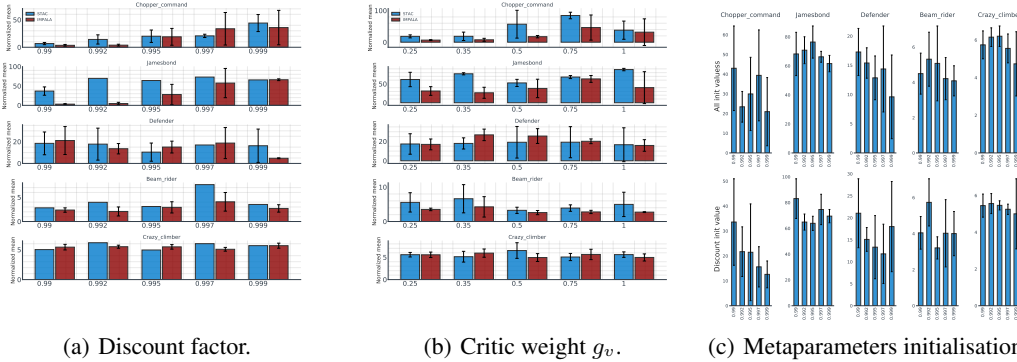

(a) Discount factor.   (b) Critic weight $g_v$.   (c) Metaparameters initialisation.

Figure 4: Robustness results in Atari. Mean and confidence intervals (over 3 seeds), after 200M frames of training. **Fig. 4(a) and Fig. 4(b):** blue bars correspond to STACX and red bars to IMPALA. Rows correspond to specific Atari games, and columns to the value of the hyper parameter in the outer loss $(\gamma, g_v)$. We observe that STACX is better than IMPALA in $72\%$ of the runs (Fig. 4(a)) and in $80\%$ of the runs in Fig. 4(b). **Fig. 4(c):** Robustness to the initialisation of the metaparameters. Columns correspond to different games. **Bottom:** perturbing $\gamma^{\text{init}} \in \{0.99, 0.992, 0.995, 0.997, 0.999\}$. **Top:** perturbing all the meta parameter initialisations. I.e., setting all the hyperparamters $\{\gamma^{\text{init}}, \lambda^{\text{init}}, g_v^{\text{init}}, g_e^{\text{init}}, g_p^{\text{init}}, \alpha^{\text{init}}\}_{i=1}^3$ to a single fixed value in $\{0.99, 0.992, 0.995, 0.997, 0.999\}$.

**Adaptivity.** In Fig. 5(a) we visualize the metaparameters of STACX during training. The metaparameters associated with the policy head (head number 1) are in blue, and the auxiliary heads (2 and 3) are in orange and magenta. We present the values of the metaparameters used in the inner loss, i.e., after we apply a sigmoid activation. But to have a single scale for all the metaparameters ($\eta \in [0, 1]$), we present the loss coefficients $g_e, g_v, g_p$ without scaling them by the respective value in the outer loss. For example, the value of the entropy weight $g_e$ that is presented in Fig. 5(a) is further multiplied by $g_e^{\text{outer}} = 0.01$ when used in the inner loss. As there are many metaparameters, seeds, and games, we only present results on a single seed (chosen arbitrarily to 1) and a single game (James Bond). In the supplementary we provide examples for all the games (Section 13).

Inspecting Fig. 5(a), one can notice that the metaparameters are being adapted in a none monotonic manner that could not have been designed by hand. We highlight a few trends which are visible in

Fig. 5(a) and we found to repeat across games (Section 13). The metaparameters of the auxiliary heads are self-tuned to have relatively similar values but different than those of the main head. For example, the main head discount factor converges to the value in the outer loss (0.995). In contrast, the auxiliary heads' discount factors often change during training and get to lower values. Another observation is that the leaky V-trace parameter $\alpha$ remains close to 1 at the beginning of training, so it is quite similar to V-trace. Towards the end of the training, it self-tunes to lower values (closer to importance sampling), consistently across games. We emphasize that these observations imply that adaptivity happens in self-tuning agents. It does not imply that this adaptivity is directly helpful. We can only deduce this connection implicitly, i.e., we observe that self-tuning agents achieve higher performance and adapt their metaparameters through training.

In Fig. 5(b), we experimented with a variation of STACX that self-tunes both $\alpha_\rho$ and $\alpha_c$ without imposing $\alpha_\rho \geq \alpha_c$ (as Theorem 1 requires to guarantee contraction). Inspecting Fig. 5(b), we can see that STACX self-tunes $\alpha_\rho \geq \alpha_c$ in James Bond. In addition, we measured that across the 57 games $\alpha_\rho \geq \alpha_c$ in 91.2% of the time (averaged over time, seeds, and games), and that $\alpha_\rho \geq 0.99\alpha_c$ in 99.2% of the time. In terms of performance, the median score (353%) was slightly worse than STACX. A possible explanation is that while this variation allows more flexibility, it may also be less stable as the contraction is not guaranteed.

In another variation we self-tuned $\alpha$ together with a single truncation parameter $\bar{\rho} = \bar{c}$. This variation performed worse, achieving a median score of 301%, which may be explained by $\bar{\rho}$ not being differentiable, suffering from nonsmooth (sub) gradients and possibly saturated IS truncation levels.

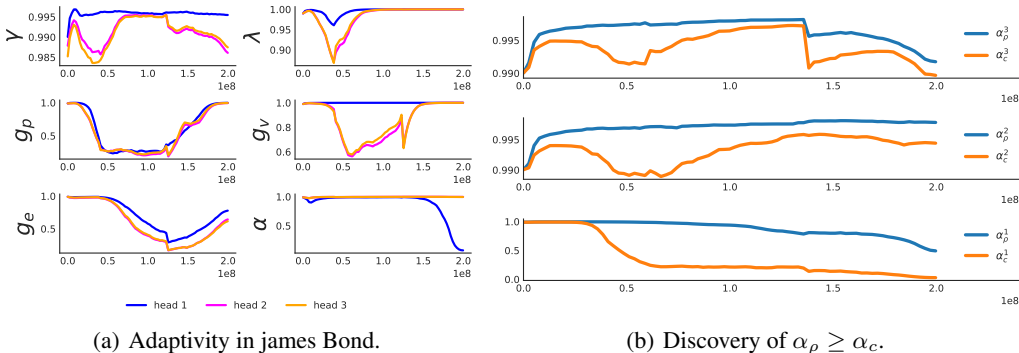

(a) Adaptivity in james Bond.　　　　(b) Discovery of $\alpha_\rho \geq \alpha_c$.

## 5　Summary

In this work, we demonstrated that it is feasible to self-tune all the differentiable hyperparameters in an actor-critic loss function. We presented STAC and STACX, actor-critic algorithms that self-tune a large number of hyperparameters of different nature (controlling important trade-offs in a reinforcement learning agent) online, within a single lifetime. We showed that these agents' performance improves as they self-tune more hyperparameters. In addition, the algorithms are computationally efficient and robust to their hyperparameters. Despite being an online algorithm, STACX achieved very high results in the ALE and the RWRL domains.

We plan to extend STACX with Experience Replay to make it more data-efficient in future work. By embracing self-tuning via metagradients, we were able to introduce these novel ideas into our agent, without having to tune their new hyperparameters. However, we emphasize that STACX is not a fully parameter-free algorithm; we hope to investigate further how to make STACX less dependent on the hyperparameters of the outer loss in future work.

# 6 Broader Impact

The last decade has seen significant improvements in Deep Reinforcement Learning algorithms. To make these algorithms more general, it became a common practice in the DRL community to measure the performance of a single DRL algorithm by evaluating it in a diverse set of environments, where at the same time, it must use a single set of hyperparameters. That way, it is less likely to overfit the agent's hyperparameters to specific domains, and more general properties can be discovered. These principles are reflected in popular DRL benchmarks like the ALE and the DM control suite.

In this paper, we focus on exactly that goal and design a self-tuning RL agent that performs well across a diverse set of environments. Our agent starts with a global loss function that is shared across the environments in each benchmark. But then, it has the flexibility to self-tune this loss function, separately in each domain. Moreover, it can adapt its loss function within a single lifetime to account for inherent non-stationarities in RL algorithms - exploration vs. exploitation, changing data distribution, and degree of off-policy.

While using meta-learning to tune hyperparameters is not new, we believe that we have made significant progress that will convince many people in the DRL community to use metagradients. We demonstrated that our agent performs significantly better than the baseline algorithm in four benchmarks. The relative improvement is much more significant than in previous metagradient papers and is demonstrated across a wider range of environments. While each of these benchmarks is diverse on its own, together, they give even more significant evidence to our approach's generality.

Furthermore, we show that it's possible to self-tune tenfold more metaparameters from different types. We also showed that we gain improvement from self-tuning various subsets of the meta parameters, and that performance kept improving as we self-tuned more metaparameters. Finally, we have demonstrated how embracing self-tuning can help to introduce new concepts (leaky V-trace and parameterized auxiliary tasks) to RL algorithms without needing tuning.

# 7 Acknowledgements

We would like to thank Adam White and Doina Precup for their comments, Manuel Kroiss for implementing the computing infrastructure described in Section 8 and Thomas Degris for the environments infrastructure design.

The author(s) received no specific funding for this work.

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
