[Supplementary Material]

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

# 8    Computing Infrastructure

We run our experiments using a distributed infrastructure implemented in JAX (Bradbury et al., 2018), using the JAX libraries Haiku (Hennigan et al., 2020), RLax (Budden et al., 2020) and Optax (Hessel et al., 2020). The computing infrastructure is based on an actor-learner decomposition (Espeholt et al., 2018), where multiple actors generate experience in parallel, and this experience is channelled into a learner.

It allows us to run experiments in two modalities. In the first modality, following Espeholt et al. (2018), the actors programs are distributed across multiple CPU machines, and both the stepping of environments and the network inference happens on CPU. The data generated by the actor programs is processed in batches by a single learner using a GPU. Alternatively, both the actors and learners are co-located on a single machine, where the host is equipped with 56 CPU cores and connected to 8 TPU cores (Jouppi et al., 2017). To minimize the effect of Python's Global Interpreter Lock, each actor-thread interacts with a *batched environment*; this is exposed to Python as a single special environment that takes a batch of actions and returns a batch of observations, but that behind the scenes steps each environment in the batch in C++. The actor threads share 2 of the 8 TPU cores (to perform inference on the network), and send batches of fixed size trajectories of length T to a queue. The learner threads takes these batches of trajectories and splits them across the remaining 6 TPU cores for computing the parameter update (these are averaged with an all reduce across the participating cores). Updated parameters are sent to the actor's TPU cores via a fast device to device channel, as soon as the new parameters are available. This minimal unit can be replicates across multiple hosts, each connected to its own 56 CPU cores and 8 TPU cores, in which case the learner updates are synced and averaged across all cores (again via fast device to device communication).

# 9    Reproducibility

## 9.1    Open-sourcing

We open-sourced our JAX implementation of the computation of leaky V-trace targets from trajectories of experience. The code can be found at www.github.com/anonymised-link.

## 9.2    Pseudo code

In this Section, we provide pseudo-code for STAC and STACX. To improve reproducibility, we also include information on where to stop gradients when computing the inner and outer losses. For that goal, we denote by $\text{sg}()$ the operation that is stopping the gradients from propagating through a function.

We begin with Algorithm 1 that explains how to compute the IMPALA loss function. Algorithm 1 gets as inputs a set of trajectories T, the agent parameters $\theta$, parameters $\zeta$ that include hyperparameters and metaparameters, and a flag that indicates if it is an inner or an outer loss (see lines 23-26 to).

At each iteration $t$ Algorithm 2 calls Algorithm 1 to update the parameters $\theta_t$ and the metaparameters $\eta_t$ by differentiating the inner loss w.r.t $\theta$ and for the outer losses w.r.t $\eta$. In line 10, given the current metaparameters $\eta_t$, we apply a set of nonlinear transformations to compute the values of the hyperparameters of the inner loss. We apply a sigmoid activation on all the metaparameters, which ensures that they remain bounded, and multiply the loss coefficients by their corresponding values in the outer loss to guarantee that they are initialized from the same values.

In line 11 the gradient of the inner loss is computed w.r.t $\theta$. Since the metaparameters define the loss, this gradient is parametrized by the metaparameters. We repeat the gradient computation for each head $p \in [1..P]$ where $P = 1$ for STAC and $P = 3$ for STACX; and the notation $\theta_t^p$ refers to the parameters that correspond to the p-th head. In practice, the heads share a single torso (see next subsection for details), so this step is computationally efficient.

In line 12 we compute the update to the parameters $\theta$ by applying an optimizer update to parameters $\theta_t$ using the gradient $\nabla \theta_t(\eta_t)$ that was compute in line 11. In practice, we use the RMSProp optimizer for that. Since the metaparameters parametrized the gradient, so does $\theta_{t+1}$.

In line 13, we differentiate the outer loss, with fixed hyperparameters $\zeta$, w.r.t the metaparameters $\eta_t$. This is done by differentiating Algorithm 1 w.r.t $\eta_t$ through the parameterized parameters $\theta_{t+1}^1(\eta_t)$.

---
**Algorithm 1** IMPALA loss with V-trace
---
1: **Inputs**: $T, \theta, \zeta$, inner loss
- **Data** T: $m$ trajectories $\{\tau_i\}_{i=1}^m$ of size $n$, $\tau_i = \left\{x_s^i, a_s^i, r_s^i, \mu(a_s^i|x_s^i)\right\}_{s=1}^n$, where $\mu(a_s^i|x_s^i)$ is the probability assigned to $a_s^i$ in state $x_s^i$ by the behaviour policy $\mu(a|x)$.
- **Hyperparameters:** $\zeta = \{g_v, g_e, g_p, \gamma, \lambda, \alpha, \bar{c}, \bar{\rho}\}$ .
- **Agent parameters:** $\theta$, which define the policy $\pi_\theta(a|x)$ and value function $V_\theta(x)$.
- **Inner Loss:** Boolean flag representing inner/outer loss.

2: **for** $i = 1, \ldots, m$ **do** $\quad\quad\quad\quad\quad\quad\quad\quad\quad\quad\quad\quad\quad\quad$ ▷ Compute leaky V-trace targets
3: $\quad$ **for** $s = 1, \ldots, n$ **do**
4: $\quad\quad$ Let $\text{IS}_s^i = \text{sg}(\frac{\pi(a_s^i|x_s^i)}{\mu(a_s^i|x_s^i)})$ $\quad\quad\quad\quad\quad\quad\quad\quad\quad$ ▷ Importance sampling ratios
5: $\quad\quad$ Set $c_s^i = \left((1-\alpha)\text{IS}_s^i + \alpha \min(\bar{c}, \text{IS}_s^i)\right)\lambda$
6: $\quad\quad$ Set $\rho_s^i = (1-\alpha)\text{IS}_s^i + \alpha \min(\bar{\rho}, \text{IS}_s^i)$
7: $\quad\quad$ Set $\delta_s^i = \rho_s^i(r_{s+1}^i + \gamma\text{sg}(V_\theta(x_{s+1}^i)) - \text{sg}(V_\theta(x_s^i)))$ $\quad\quad\quad\quad$ ▷ One step td errors
8: $\quad$ **end for**
9: $\quad$ Let $e_{n+1}^i = 0$
10: $\quad$ **for** $s = n, \ldots, 1$ **do** $\quad\quad\quad\quad\quad\quad\quad\quad\quad$ ▷ Compute n-step td-errors backwards
11: $\quad\quad e_s^i = \delta_s^i + \gamma c_s^i e_{s+1}^i$
12: $\quad$ **end for**
13: $\quad$ **for** $s = 1, \ldots, n$ **do** $\quad\quad\quad\quad\quad\quad\quad\quad\quad\quad$ ▷ Compute leaky V-trace targets
14: $\quad\quad v_s^i = e_s^i + \text{sg}(V_\theta(x_s^i))$
15: $\quad$ **end for**
16: **end for**

17: $L_{\text{Value}}(\theta) = g_v \cdot \sum_{i=1,s=1}^{i=m,s=n-1}\left(v_s^i - V_\theta(x_s^i)\right)^2$
18: $L_{\text{Entropy}}(\theta) = -g_e \cdot \sum_{i=1,s=1}^{i=m,s=n-1}\sum_a \pi(a_s^i|x_s^i)\log(\pi(a_s^i|x_s^i))$
19: **if** Inner Loss **then**
20: $\quad L_{\text{Policy}}(\theta) = -g_p \cdot \sum_{i=1,s=1}^{i=m,s=n-1}\rho_s^i \log(\pi(a_s^i|x_s^i))\left(r_{s+1}^i + \gamma v_{s+1}^i - \text{sg}(V_\theta(x_s^i))\right)$
21: **else**
22: $\quad L_{\text{Policy}}(\theta) = -g_p \cdot \sum_{i=1,s=1}^{i=m,s=n-1}\rho_s^i \log(\pi(a_s^i|x_s^i))\text{sg}\left(r_{s+1}^i + \gamma v_{s+1}^i - V_\theta(x_s^i)\right)$
23: **end if**
24: **Return** $L(\theta) = L_{\text{Value}}(\theta) + L_{\text{Policy}}(\theta) + L_{\text{Entropy}}(\theta)$

---
**Algorithm 2** Inner and outer loss
---
1: Let $\text{Loss}(T, \theta, \zeta, \text{inner loss})$ be a function that calls Algorithm 1.
2: Let $\sigma(x)$ denote applying a sigmoid activation on $x$.
3: Let $\bar{\rho} = 1, \bar{c} = 1$ be fixed truncation levels.
4: Denote the metaparameters at time $t$ by $\eta_t = \{\gamma, \lambda, \alpha, g_v, g_p, g_e\}$
5: Denote the hyperparameters by $\zeta = \left\{\gamma^{\text{outer}}, \lambda^{\text{outer}}, \alpha^{\text{outer}}, g_v^{\text{outer}}, g_p^{\text{outer}}, g_e^{\text{outer}}, \bar{\rho}, \bar{c}\right\}$
6: Let OPT, MetaOPT be the optimizer and meta optimizer with their respective hyper parameters
7: Let $g^{\text{KL}}$ be the KL loss coefficient.
8: **for** t = 1 ... **do**
9: $\quad$ Collect trajectories $T = \{\tau_i\}_{i=1}^m$ using the behaviour policy $\mu_t$
10: $\quad$ Set $\zeta(\eta_t) = \left\{\sigma(\gamma), \sigma(\lambda), \sigma(\alpha), \sigma(g_v)g_v^{\text{outer}}, \sigma(g_p)g_p^{\text{outer}}, \sigma(g_e)g_e^{\text{outer}}, \bar{\rho}, \bar{c}\right\}$
11: $\quad \nabla\theta_t(\eta_t) = \frac{1}{P}\sum_{p=1}^P \nabla_\theta\left(\text{Loss}(T, \theta_t^p, \zeta(\eta_t), \text{True})\right)$ $\quad\quad$ ▷ Gradient of the inner loss w.r.t $\theta$
12: $\quad \theta_{t+1}(\eta_t) = \text{OPT}(\theta_t, \nabla\theta_t(\eta_t))$
13: $\quad \nabla\eta_t = \nabla_\eta\left(\text{Loss}(T, \theta_{t+1}^1(\eta_t), \zeta, \text{False})\right)$ $\quad\quad\quad\quad$ ▷ Metagradient of the outer loss w.r.t $\eta$
14: $\quad \nabla\eta_t = \nabla\eta_t + g^{\text{KL}}\nabla_{\eta_t}\text{KL}(\pi_{\theta_{t+1}(\eta_t)}, \pi_{\theta_t}; T)$ $\quad\quad\quad$ ▷ Metagradient of the KL loss
15: $\quad \eta_{t+1} = \text{MetaOPT}(\eta_t, \nabla\eta_t)$
16: **end for**

Notice that when we compute the outer loss, we only take into account the loss of head 1. In line 14, we update the meta parameters by calling the meta optimizer (ADAM with default values).

## 9.3 Hyperparameters

**Architectures.**

Table 2: Network architectures

| Parameter | Atari | Control - pixels | Control - features |
|---|---|---|---|
| convolutions in block | (2, 2, 2) | (2, 2, 2) | - |
| channels | (13, 32, 32) | (32, 32, 32) | - |
| kernel sizes | (3, 3, 3) | (3, 3, 3) | - |
| kernel strides | (1, 1, 1) | (2, 2, 2) | |
| pool sizes | (3, 3, 3) | - | - |
| pool strides | (2, 2, 2) | - | - |
| mlp torso | - | - | (256, 256) |
| lstm | - | 256 | 256 |
| frame stacking | 4 | - | - |
| head hiddens | 256 | 256 | 256 |
| activation | Relu | Relu | Relu |

Our DNN architecture is composed of a shared torso, which then splits to different heads. We have a head for the policy and a head for the value function (multiplied by three for STACX). Each head is a two-layered MLP with 256 hidden units, where the output dimension corresponds to 1 for the vale function head. For the policy head, we have $|A|$ outputs that correspond to softmax logits when working with discrete actions (Atari), and $2|A|$ outputs that correspond to the mean and standard deviation of a squashed Gaussian distribution with a diagonal covariance matrix in the case of continuous actions. We use ReLU activations on the outputs of all the layers besides the last layer.

We use a softmax distribution for the policy and the entropy of this distribution for regularization for discrete actions. For continuous actions, we apply a tanh activation on the output of the mean. For the standard deviation, for output $y$, the standard deviation is given by

$$\sigma(y) = \exp \sigma_{\min} + 0.5 \cdot (\sigma_{\max} - \sigma_{\min}) \cdot (\tanh(y) + 1))$$

We then sample action from a Gaussian distribution with a diagonal covariance matrix $N(\mu, \sigma)$ and apply a tanh activation on the sample. These transformations guarantee that the sampled action is bounded. We then adjust the probability and log probabilities of the distribution by making a Jacobian correction (Haarnoja et al., 2018).

The **torso** of the network is defined per domain. When learning from features (RWRL and DM control), we use a two-layered MLP, with hidden layers specified in Table 2. For Atari and DM control from pixels, our network uses a convolution torso. The torso is composed of residual blocks. In each block there is a convolution layer, with stride, kernel size, channels specified per domain (Atari, Control from pixels) in Table 2, with an optional pooling layer following it. The convolution layer is followed by n - layers of convolutions (specified by blocks), with a skip contention. The output of these layers is of the same size of the input so they can be summed. The block convolutions have kernel size 3, stride 1. The torso is followed by an LSTM (size specified in Table 2). In Atari, we do not use an LSTM, but use frame stacking (4) instead.

**Hyperparameters.** Table 3 lists all the hyperparameters used by STAC and STACX. Most of the hyperparameters are shared across all domains (listed by Value), and follow the reported parameters from the IMPALA paper. Whenever they are not, we list the specific values that are used in each domain (listed by domain).

As a design choice, we did not tune many of the hyperparameters. Instead, we use the hyperparameters that were reported for IMPALA in earlier work. For new hyperparameters, we preferred using default values.

Table 3: Hyperparameters table

| Parameter | Value | Atari | Control |
|---|---|---|---|
| total envirnoment steps | - | 200e6 | 30e6 |
| optimizer | RMSPROP | - | - |
| start learning rate | - | $6 \cdot 10^{-4}$ | $10^{-3}$ |
| end learning rate | - | 0 | $10^{-4}$ |
| decay | 0.99 | - | - |
| eps | 0.1 | - | - |
| batch size (m) | - | 32 | 24 |
| trajectory length (n) | - | 20 | 40 |
| overlap length | - | 0 | 30 |
| $\gamma^{\text{outer}}$ | - | 0.995 | 0.99 |
| $\lambda^{\text{outer}}$ | 1 | - | - |
| $\alpha^{\text{outer}}$ | 1 | - | - |
| $g_e^{\text{outer}}$ | 0.01 | - | - |
| $g_v^{\text{outer}}$ | 0.25 | - | - |
| $g_p^{\text{outer}}$ | 1 | - | - |
| $g^{\text{kl}}$ | 1 | - | - |
| meta optimizer | Adam | - | - |
| meta learning rate | $10^{-3}$ | - | - |
| b1 | 0.9 | - | - |
| b2 | 0.999 | - | - |
| eps | 1e-4 | - | - |
| $\eta^{\text{init}}$ | 4.6 | - | - |

For example, we chose Adam as a meta optimizer with default hyperparameters that were not tuned. For control, we use $\gamma = 0.99$, which is the default value of many agents in this domain. The network architectures are quite standard as well and was used in the IMPALA paper.

We now list a few things that we did try to tune.

**Number of auxiliary tasks.** We have also experimented with other amounts of auxiliary losses, e.g., having $2, 5$, or $8$ auxiliary loss functions. In Atari, these variations performed better than having a single loss function (STAC) but slightly worse than having 3. This can be further explained by Fig. 5(a), which shows that the auxiliary heads are self-tuned to similar metaparameters.

**Behaviour policy.** We considered two variations of STACX that allow the other heads to act.

1. Random ensemble: The policy head is chosen at random from $[1, .., n]$,, and the hyperparameters are differentiated w.r.t the performance of each of the heads in the outer loop.

2. Average ensemble: The actor policy is defined to be the average logits of the heads, and we learn one additional head for the value function of this policy.

The metagradient in the outer loop is taken w.r.t the actor policy, and /or, each one of the heads individually. While these extensions seem interesting, in all of our experiments, they always led to a small decrease in performance when compared to our auxiliary task agent without these extensions. Similar findings were reported in (Fedus et al., 2019).

**KL coefficient**. When we introduced this loss function, we did not use a coefficient for it. Thus, it had the default value 1. Later on, we revisited this design choice and tested what happens when we use $g^{\text{KL}} \in \{0, 0.3, 1, 2\}$. We observed that the value of this parameter could affect our results, but not significantly, and we chose to remain with the default value (1), which seemed to perform the best.

### 9.4 Resource Usage

The average run time in the different environments is reported in Table 4. Thanks to massively parallelism, with modern hardware such as TPUs, agents are often bottlenecked by data in-feed rather than actual computation. This is true despite the use of fairly deep networks. As a result, we can increase the exact amount of computation performed on each step with a modest impact on the

Table 4: Run times in minutes

| Domain | IMPALA | STAC | IMPALA Aux | STACX |
|---|---|---|---|---|
| RWRL | 44 | 44.3 | 55.3 | 56 |
| DM control, features | 30.1 | 30.3 | 36.8 | 36.9 |
| DM control, pixels | 93 | 93 | 97 | 97 |
| Atari | 70 | 71 | 83 | 84 |

runtime. Inspecting Table 4, we can see that self-tuning indeed results with a very mild increase in run time. However, this does not mean that self-tuning costs the same amount of compute, which is hard to measure.

To further investigate this, we measured the run time in Atari with a second hardware configuration (the combination of distributed CPUs with a GPU learner from section 8). The run times of the different agents in Atari were 105, 129, 106, and 133 (for IMPALA, STAC, IMPALA Aux and STACX respectively). With this hardware, the run time of all the agents was longer. When we compare the run time of the different agents, we can see that self-tuning required about $25\%$ more time, while the extra run time from having auxiliary loss functions was negligible.

To conclude, the run times of the different agents is hardware specific. We observed that overall STAC and STACX result in slightly more compute than the baseline agent.

### 9.5 Reproducing Baseline Algorithms

Inspecting the results in Fig. 2(b), one may notice small differences between the results of IMPALA and using meta gradients to tune only $\lambda, \gamma$ compared to the results that were reported in (Xu et al., 2018).

We investigated the possible reasons for these differences. First, our method was implemented in a different codebase. Our code is written in JAX, compared to the implementation in (Xu et al., 2018) that was written in TensorFlow. This may explain the small difference in final performance between our IMPALA baseline that achieved $243\%$ median score (Fig. 2(b)) and the result of Xu et al., which is slightly higher ($257.1\%$).

Second, Xu et al. observed that embedding the $\gamma$ hyperparameter into the $\theta$ network improved their results significantly, reaching a final performance (when learning $\gamma, \lambda$) of $287.7\%$ when self-tuning only $\gamma$ and $\lambda$ (see section 1.4 in (Xu et al., 2018) for more details). When tuning only $\gamma$, Xu et al. (2018) report that without using the embedding, the performance of their meta gradient agent drops to $183\%$, even below the IMPALA baseline, where when using the $\gamma$ embedding, the performance increases to $267.9\%$ (for self-tuning only $\gamma$).

When self-tuning only $\gamma$ but without using embedding, we also observed a slight decrease in performance ($243\%$->$240\%$), but not as significant as in (Xu et al., 2018). We further investigated this difference by introducing the $\gamma$ embedding into our architecture. With $\gamma$ embedding, our method achieved a score of $280.6\%$ (for self tuning only $\lambda, \gamma$), which almost reproduces the results in (Xu et al., 2018).

We also introduced the same embedding mechanism to STACX when self-tuning all the metaparameters. In this case, for auxiliary loss $i$ we embed $\gamma_i$. We experimented with two variants, one that shares the embedding weights across the auxiliary tasks and one that learns a specific embedding for each auxiliary task. Both of these variants performed similarly ($306.8\%$, $307.7\%$ respectively), which is better than the result for self-tuning only $\gamma, \lambda$ ($280.6\%$). However, STACX performed better without the embedding ($364\%$), so we did not use the $\gamma$ embedding in our architecture. We leave it to future work to further investigate methods of combining the embedding mechanisms with the auxiliary loss functions.

## 10 Analysis of Leaky V-trace

Define the Leaky V-trace operator $\tilde{\mathcal{R}}$:

$$\tilde{\mathcal{R}}V(x) = V(x) + \mathbb{E}_\mu \left[ \sum_{t \geq 0} \gamma^t \left( \Pi_{i=0}^{t-1} \tilde{c}_i \right) \tilde{\rho}_t \left( r_t + \gamma V(x_{t+1}) - V(x_t) \right) | x_0 = x, \mu \right], \quad (6)$$

where the expectation $\mathbb{E}_\mu$ is with respect to the behaviour policy $\mu$ which has generated the trajectory $(x_t)_{t \geq 0}$, i.e., $x_0 = x, x_{t+1} \sim P(\cdot | x_t, a_t), a_t \sim \mu(\cdot | x_t)$. Similar to (Espeholt et al., 2018), we consider the infinite-horizon operator but very similar results hold for the n-step truncated operator.

Let

$$\text{IS}(x_t) = \frac{\pi(a_t | x_t)}{\mu(a_t | x_t)},$$

be importance sampling weights, let

$$\rho_t = \min(\bar{\rho}, \text{IS}(x_t)), \quad c_t = \min(\bar{c}, \text{IS}(x_t)),$$

be truncated importance sampling weights with $\bar{\rho} \geq \bar{c}$, and let

$$\tilde{\rho}_t = \alpha_\rho \rho_t + (1 - \alpha_\rho)\text{IS}(x_t), \quad \tilde{c}_t = \alpha_c c_t + (1 - \alpha_c)\text{IS}(x_t)$$

be the Leaky importance sampling weights with leaky coefficients $\alpha_\rho \geq \alpha_c$.

**Theorem 2** (Restatement of Theorem 1). *Assume that there exists $\beta \in (0, 1]$ such that $\mathbb{E}_\mu \rho_0 \geq \beta$. Then the operator $\tilde{\mathcal{R}}$ defined by Eq. (6) has a unique fixed point $\tilde{V}^{\tilde{\pi}}$, which is the value function of the policy $\pi_{\bar{\rho}, \alpha_\rho}$ defined by*

$$\pi_{\bar{\rho}, \alpha_\rho} = \frac{\alpha_\rho \min \left( \bar{\rho}\mu(a|x), \pi(a|x) \right) + (1 - \alpha_\rho)\pi(a|x)}{\sum_b \alpha_\rho \min \left( \bar{\rho}\mu(b|x), \pi(b|x) \right) + (1 - \alpha_\rho)\pi(b|x)},$$

*Furthermore, $\tilde{\mathcal{R}}$ is a $\tilde{\eta}$-contraction mapping in sup-norm, with*

$$\tilde{\eta} = \gamma^{-1} - (\gamma^{-1} - 1)\mathbb{E}_\mu \left[ \sum_{t \geq 0} \gamma^t \left( \Pi_{i=0}^{t-2} \tilde{c}_i \right) \tilde{\rho}_{t-1} \right] \leq 1 - (1 - \gamma)(\alpha_\rho \beta + 1 - \alpha_\rho) < 1,$$

*where $\tilde{c}_{-1} = 1, \tilde{\rho}_{-1} = 1$ and $\Pi_{s=0}^{t-2} c_s = 1$ for $t = 0, 1$.*

*Proof.* The proof follows the proof of V-trace from (Espeholt et al., 2018) with adaptations for the leaky V-trace coefficients. We have that

$$\tilde{\mathcal{R}}V_1(x) - \tilde{\mathcal{R}}V_2(x) = \mathbb{E}_\mu \sum_{t \geq 0} \gamma^t \left( \Pi_{s=0}^{t-2} c_s \right) \left[ \tilde{\rho}_{t-1} - \tilde{c}_{t-1}\tilde{\rho}_t \right] \left( V_1(x_t) - V_2(x_t) \right).$$

Denote by $\tilde{\kappa}_t = \rho_{t-1} - \tilde{c}_{t-1}\tilde{\rho}_t$, and notice that

$$\mathbb{E}_\mu \tilde{\rho}_t = \alpha_\rho \mathbb{E}_\mu \rho_t + (1 - \alpha_\rho)\mathbb{E}_\mu \text{IS}(x_t) \leq 1,$$

since $\mathbb{E}_\mu \text{IS}(x_t) = 1$, and therefore, $\mathbb{E}_\mu \rho_t \leq 1$. Furthermore, since $\bar{\rho} \geq \bar{c}$ and $\alpha_\rho \geq \alpha_c$, we have that $\forall t, \tilde{\rho}_t \geq \tilde{c}_t$. Thus, the coefficients $\tilde{\kappa}_t$ are non negative in expectation, since

$$\mathbb{E}_\mu \tilde{\kappa}_t = \mathbb{E}_\mu \tilde{\rho}_{t-1} - \tilde{c}_{t-1}\tilde{\rho}_t \geq \mathbb{E}_\mu \tilde{c}_{t-1}(1 - \tilde{\rho}_t) \geq 0.$$

Thus, $V_1(x) - V_2(x)$ is a linear combination of the values $V_1 - V_2$ at the other states, weighted by non-negative coefficients whose sum is

$$\sum_{t\geq 0}\gamma^t\mathbb{E}_\mu\left(\Pi_{s=0}^{t-2}\tilde{c}_s\right)[\tilde{\rho}_{t-1}-\tilde{c}_{t-1}\tilde{\rho}_t]$$

$$=\sum_{t\geq 0}\gamma^t\mathbb{E}_\mu\left(\Pi_{s=0}^{t-2}\tilde{c}_s\right)\tilde{\rho}_{t-1}-\sum_{t\geq 0}\gamma^t\mathbb{E}_\mu\left(\Pi_{s=0}^{t-1}\tilde{c}_s\right)\tilde{\rho}_t$$

$$=\gamma^{-1}-(\gamma^{-1}-1)\sum_{t\geq 0}\gamma^t\mathbb{E}_\mu\left(\Pi_{s=0}^{t-2}\tilde{c}_s\right)\tilde{\rho}_{t-1}$$

$$\leq\gamma^{-1}-(\gamma^{-1}-1)(1+\gamma\mathbb{E}_\mu\tilde{\rho}_0) \tag{7}$$

$$=1-(1-\gamma)\mathbb{E}_\mu\tilde{\rho}_0$$

$$=1-(1-\gamma)\mathbb{E}_\mu\left(\alpha_\rho\rho_0+(1-\alpha_\rho)\mathrm{IS}(x_0)\right)$$

$$\leq 1-(1-\gamma)\left(\alpha_\rho\beta+1-\alpha_\rho\right)<1, \tag{8}$$

where Eq. (7) holds since we expanded only the first two elements in the sum, and all the elements in this sum are positive, and Eq. (8) holds by the assumption.

We deduce that $\|\tilde{\mathcal{R}}V_1(x)-\tilde{\mathcal{R}}V_2(x)\|_\infty\leq\tilde{\eta}\|V_1(x)-V_2(x)\|_\infty$, with $\tilde{\eta}=1-(1-\gamma)(\alpha_\rho\beta+1-\alpha_\rho)<1$, so $\tilde{\mathcal{R}}$ is a contraction mapping. Furthermore, we can see that the parameter $\alpha_\rho$ controls the contraction rate, for $\alpha_\rho=1$ we get the contraction rate of V-trace $\tilde{\eta}=1-(1-\gamma)\beta$ and as $\alpha_\rho$ gets smaller with get better contraction as with $\alpha_\rho=0$ we get that $\tilde{\eta}=\gamma$.

Thus $\tilde{\mathcal{R}}$ possesses a unique fixed point. Let us now prove that this fixed point is $V^{\pi_{\bar{\rho},\alpha_\rho}}$, where

$$\pi_{\bar{\rho},\alpha_\rho}=\frac{\alpha_\rho\min\left(\bar{\rho}\mu(a|x),\pi(a|x)\right)+(1-\alpha_\rho)\pi(a|x)}{\sum_b\alpha_\rho\min\left(\bar{\rho}\mu(b|x),\pi(b|x)\right)+(1-\alpha_\rho)\pi(b|x)}, \tag{9}$$

is a policy that mixes the target policy with the V-trace policy.

We have:

$$\mathbb{E}_\mu\left[\tilde{\rho}_t(r_t+\gamma V^{\pi_{\bar{\rho},\alpha_\rho}}(x_{t+1})-V^{\pi_{\bar{\rho},\alpha_\rho}}(x_t))|x_t\right]$$

$$=\mathbb{E}_\mu\left(\alpha_\rho\rho_t+(1-\alpha_\rho)\mathrm{IS}(x_t)\right)\left(r_t+\gamma V^{\pi_{\bar{\rho},\alpha_\rho}}(x_{t+1})-V^{\pi_{\bar{\rho},\alpha_\rho}}(x_t)\right)$$

$$=\sum_a\mu(a|x_t)\left(\alpha_\rho\min\left(\bar{\rho},\frac{\pi(a|x_t)}{\mu(a|x_t)}\right)+(1-\alpha_\rho)\frac{\pi(a|x_t)}{\mu(a|x_t)}\right)\left(r_t+\gamma V^{\pi_{\bar{\rho},\alpha_\rho}}(x_{t+1})-V^{\pi_{\bar{\rho},\alpha_\rho}}(x_t)\right)$$

$$=\sum_a\left(\alpha_\rho\min\left(\bar{\rho}\mu(a|x_t),\pi(a|x_t)\right)+(1-\alpha_\rho)\pi(a|x_t)\right)\left(r_t+\gamma V^{\pi_{\bar{\rho},\alpha_\rho}}(x_{t+1})-V^{\pi_{\bar{\rho},\alpha_\rho}}(x_t)\right)$$

$$=\sum_a\pi_{\bar{\rho},\alpha_\rho}(a|x_t)(r_t+\gamma V^{\pi_{\bar{\rho},\alpha_\rho}}(x_{t+1})-V^{\pi_{\bar{\rho},\alpha_\rho}}(x_t))\cdot\sum_b\left(\alpha_\rho\min\left(\bar{\rho}\mu(b|x_t),\pi(b|x_t)\right)+(1-\alpha_\rho)\pi(b|x_t)\right)=0,$$

where we get that the left side (up to the summation on $b$) of the last equality equals zero since this is the Bellman equation for $V^{\pi_{\bar{\rho},\alpha_\rho}}$. We deduce that $\tilde{\mathcal{R}}V^{\pi_{\bar{\rho},\alpha_\rho}}=V^{\pi_{\bar{\rho},\alpha_\rho}}$, thus, $V^{\pi_{\bar{\rho},\alpha_\rho}}$ is the unique fixed point of $\tilde{\mathcal{R}}$. $\qquad\square$

# 11 Individual domain learning curves in DM control and RWRL

Figure 5: Learning curves in specific domains of the RWRL challenge (Dulac-Arnold et al., 2020). In each domain we report the mean, averaged over 3 seeds, of each method with std confidence intervals as shaded areas. In addition, we report the scores for two baselines (D4PG, DMPO) at the end of training, taken from (Dulac-Arnold et al., 2020).

Figure 6: Learning curves in specific domains of the DM control suite (Tassa et al., 2018) using the features. In each domain we report the mean, averaged over 3 seeds, of each method with std confidence intervals as shaded areas. In addition, we report the scores of the A3C baseline at the end of training, taken from (Tassa et al., 2018).

Figure 7: Learning curves in specific domains of the DM control suite (Tassa et al., 2018) using pixels. In each domain we report the mean, averaged over 3 seeds, of each method with std confidence intervals as shaded areas.

# 12 Relative improvement in percents of STACX and STAC over IMPALA in Atari

Figure 8: Mean human-normalized scores after 200M frames, relative improvement in percents of STACX over IMPALA.

Figure 9: Mean human-normalized scores after 200M frames, relative improvement in percents of STAC over the IMPALA baseline.

# 13 Individual game learning curves

For clarity, we repeat a comment from the main paper regarding the values of the metaparameters. We present the values of the metaparameters used in the inner loss, i.e., after we apply a sigmoid activation. But to have a single scale for all the metaparameters ($\eta \in [0, 1]$), we present the loss coefficients $g_e, g_v, g_p$ without scaling them by the respective value in the outer loss. For example, the value of the entropy weight $g_e$ is further multiplied by $g_e^{\text{outer}} = 0.01$ when used in the inner loss.

Figure 10: Meta parameters and reward in each Atari game (and seed) during learning. Different colors correspond to different heads, blue is the main (policy) head.

Figure 11: Meta parameters and reward in each Atari game (and seed) during learning. Different colors correspond to different heads, blue is the main (policy) head.

Figure 12: Meta parameters and reward in each Atari game (and seed) during learning. Different colors correspond to different heads, blue is the main (policy) head.

Figure 13: Meta parameters and reward in each Atari game (and seed) during learning. Different colors correspond to different heads, blue is the main (policy) head.

Figure 14: Meta parameters and reward in each Atari game (and seed) during learning. Different colors correspond to different heads, blue is the main (policy) head.

Figure 15: Meta parameters and reward in each Atari game (and seed) during learning. Different colors correspond to different heads, blue is the main (policy) head.

Figure 16: Meta parameters and reward in each Atari game (and seed) during learning. Different colors correspond to different heads, blue is the main (policy) head.

Figure 17: Meta parameters and reward in each Atari game (and seed) during learning. Different colors correspond to different heads, blue is the main (policy) head.

Figure 18: Meta parameters and reward in each Atari game (and seed) during learning. Different colors correspond to different heads, blue is the main (policy) head.

Figure 19: Meta parameters and reward in each Atari game (and seed) during learning. Different colors correspond to different heads, blue is the main (policy) head.