[Reviews · NeurIPS 2020]

Review 1

Summary and Contributions: This paper proposes a STAC to self tune all the differentible hyperparameters in loss function, and also introduces a novel leaky V-trace operator. On top of these, it introduces auxiliary loss functions to enable self-tuning of the discount factors, leading to a new agent STACX. The paper conducts extensive experiments for STAC and STACX, in atari domain, DM control suite and real-world RL challenge. It shows that STAC consistently improve performance when increasing the number of self-tuning metaparameters. Also it demonstrates the trend of self-tuned metaparameters and robustness to its hyperparameters.

Strengths: 1. This paper proposes a self-tuning actor critic algorithm, using the metagradient to update all differentiable hyperparameters. It proposes a variant of V-trace operator and auxiliary loss functions to allow more hyperparameters to be differentiable. Also, it provides a theorectical guarantee of contration for the leack V-trace operator. 2. I think the main contribution in this paper is the empirical results of STAC. The proposed method was evaluated on multiple domains and demonstrates its effitiveness and significantly ourperformances baseline methods. This sheds light on the using of metagradients in many other RL topics. I like this paper in general.

Weaknesses: My major concern is the novelty about the self-tuning technique in this paper. The way STAC self-tunes the metaparameters by metagradients is straightforward --- introducing a inner loss that parameterized by metaparameters and differentiate the outer loss w.r.t the metaparameters. This is standard way in meta-learning field. Although STAC proposes a novel V-trace operator and auxiliary loss functions, these two contributions only try to make some hyperparameter differentiable for STAC, which are a little bit adhoc. I'd like to see some deep insights from the authors about the self-tuning (metagradients) itself. Other comments: 1. The paper introduces self-tuning metaparameters in the inner loss function. However, the metaparameters are bounded in (0,1) and multiplied by outer prior hyperparameters (g_v^outer, \gamma) which have been well-tuned by other algorithms. Althougn this prior can guarantee the agent is not misleaded by some very large metaparameters, I think this is somehow restricted since the metaparameters are only learned to self-tune the scale of other well-tuned hyperparameters and it can potentially limit the ability of metaparameters. Besides, there is still some hyperparameters in outer loss function. Can we make them differentiable? 2. The paper introduces auxiliary loss functions to enable the self-tuning of discount factor. Is it necessary at all? Have you tried just differentiating the inner loss function w.r.t. the discount factor? At the moment I am not sure which is a better way. 3. It seems that in Atari domain, STAC only slightly surpasses baselines in the later training stage. But in other domains, STAC achieves much better sample efficiency than baselines. Could you explain it a little bit more? 4. Can the self-tuning idea be adopted to off-policy Q-learning? DQN-based algorithms seem more sensitive to hyperparameters than A2C, such as replay buffer size.

Correctness: Yes.

Clarity: The paper is well written.

Relation to Prior Work: Yes.

Reproducibility: Yes

Additional Feedback: ** After reading response ** I think the paper would be interesting to researchers at NeurlPS since it demonstrates the power of self-tuning and perhaps the idea could be used in a wide range of of RL algorithms.


Review 2

Summary and Contributions: The paper proposes to use meta-gradients to adapt the hyperparameters of a modified IMPALA algorithm. The modifications include a version of V-trace that interpolates between truncated importance sampling and ordinary importance sampling, and additional auxiliary tasks based on learning different discount factors. The paper shows improved performance on the Arcade Learning Environment and DeepMind Control suite.

Strengths: The paper is well-written, and attempts to address a very important research question: how can the huge (and growing) list of hyperparameters in deep RL methods be set in a way that avoids expensive and brittle hyperparameter tuning? For this reason the paper seems very relevant to the NeurIPS community, and despite some issues the empirical results look somewhat promising.

Weaknesses: The proposed modifications seem heuristically motivated (and not theoretically justified), with many design decisions made that were not explained or carefully studied. Why a KL divergence penalty in the outer loop to prevent policy changes? Why leaky v-trace over other policy evaluation algorithms that don't clip importance sampling ratios? Why leaky v-trace in the inner loss but not the outer loss? Why add auxiliary tasks to a paper about using meta-gradients to adapt hyperparameters? Why auxiliary tasks solely built around discount factors? The implied justification for these (and other) design decisions seems to be improved performance on the ALE and DM control suite. However, the results are averaged over only 3 random seeds, which makes it impossible to draw conclusions, and alternatives to the design decisions weren't compared. The paper seems to contain two separate ideas: auxiliary tasks based on different discount rates, and using meta-gradients to adapt hyperparameters. It seems like there's not enough space for an in-depth, careful investigation of both.

Correctness: The empirical methodology is weak: only 3 random seeds, and very few alternative design decisions were compared.

Clarity: The paper was fairly clearly written, and did a good job of summarizing IMPALA and the proposed modifications. The paper was very clear on what was done, but not always clear on why.

Relation to Prior Work: The paper clearly discussed how it differs from the closest prior work of Xu et al. (2018), but does not mention—nor compare against—existing methods to adapt specific hyperparameters like the step size or trace decay rate.

Reproducibility: Yes

Additional Feedback: After reading the other reviews and the author response, I have decided to increase my score. I think the main contribution of demonstrating the effectiveness of self-tuning hyperparameters is probably significant enough and interesting enough to the community to warrant publication, despite the issues raised by the reviewers. The paper states multiple times that STACX "reasons about multiple horizons", but there's no explicit reasoning going on; the agent is simply minimizing auxiliary losses with different discount factors that share a layer of weights. This seems like an unwarranted anthropomorphism, and I'd recommend removing these statements. If the bias introduced by v-trace's clipping of importance weights is undesirable—implied by leaky v-trace partially undoing the clipping—why not avoid the clipping in the first place? Why not consider other solutions to the problem or other off-policy policy evaluation algorithms that don't introduce bias by clipping? Averaging results over 3 seeds is not enough to say anything about statistical significance of results or draw meaningful conclusions. Similarly, plotting half of a standard deviation is misleading: non-overlapping fractions of a standard deviation communicates nothing about statistical significance (although in some experiments the .5 standard deviation actually does overlap), yet gives the illusion of statistical significance to an unwary reader who mistakes them for confidence intervals. It would be better to plot confidence intervals and do enough runs to make them not overlap, which visually communicates to the reader that the difference in performance is statistically significant. If the required number of runs to attain statistical significance can't be done due to computational cost, I would recommend moving to a smaller environment where the proposed method can be studied more thoroughly. After that the methods can be scaled up to demonstrate their effectiveness on the ALE or DM control suite. This has the added benefit that the method being developed won't "overfit" to one environment. The auxiliary tasks seemed to harm performance or not make a difference (the pixel observations results have overlapping .5 std dev regions) on the DM control suite experiments while performing well on the ALE. The discussion section explains this away without actually showing any evidence to support the explanations. Overall, I really like the idea of using meta-gradients to adapt hyperparameters, and I think this method has a lot of promise. It would have been much better in my opinion if the paper had more thoroughly explored hyperparameter adaptation via meta-gradients on a smaller domain, then scaled the ideas up to the ALE and/or DM control suite. It would also be better to separate the auxiliary task idea into another paper; it seems orthogonal to hyperparameter adaptation via meta-gradients, and the space could've been used to more thoroughly investigate the idea of hyperparameter adaptation via meta-gradients. I also appreciate the investigation into hyperparameter robustness in section 5, although again the few random seeds makes it difficult to conclude anything from the results. Overall I think the paper has enough flaws that I'm hesitant to recommend it for publication. However, if the authors sufficiently address the questions and concerns raised in my review (and the other reviews), then I will consider raising my score.


Review 3

Summary and Contributions: The paper proposes a meta-gradient method to self-tune the differentiable parameters of an actor-critic algorithm IMPALA. They also add auxillary loass-functions in order to reason about multiple horizons, useful for learning from limited amount of data.

Strengths: *Tuning hyper-parameters is a tedious task in DRL approaches. While the idea of using meta-gradients is not new, self-tuning hyper-parameters using meta-gradients for IMPALA is a relevant contribution. *The paper implements leaky-V Trace (for STAC) with a theoretical proof of its convergence properties * STAC with auxillary tasks is implemented by introducing additional heads with the need to learn additional meta-parameters. *The experimental evaluation is extensive. Experiments are conducted on Atari as well as DM Control suite and the results verify the performance gains in using self-tuning. Robustness experiments have also been performed.

Weaknesses: *Since the paper applies the existing meta-gradients technique towards IMPALA, the novelty of the work is limited. In fact, Xu et al, 2018 applied meta-gradient to tune lambda and gamma for IMPALA. *Since meta-gradients is applied only towards a specific actor-critic framework (IMPALA), the contributions can be seen as an incremental one. *Fig 4 should be STACX Vs IMPALA, wrong legend? *In Fig 3, were the hyper-parmaters for IMPALA experiements fine tuned? Are they the best values? *The run-time of STAC and STACX is 25% more than IMPALA. How will you justify the trade-off between performance gains and run-time?

Correctness: yes, the claims seem to be correct. Theoretical proofs and empirical evaluations verify the claims.

Clarity: yes, the paper is clearly written

Relation to Prior Work: yes, relevant related literature have been discussed and compared

Reproducibility: Yes

Additional Feedback:

[Author Response · NeurIPS 2020]

We would like to thank the reviewers for their comments. Reading the reviews, we got the impression that the reviewers enjoyed certain parts in our paper but had concerns about other parts. We hope that our rebuttal addresses these concerns and would like to encourage the reviewers to consider changing their scores if this is the case.

**Novelty.** RL research suffers fundamentally from a major practical research bottleneck - the ability to set hyperparameters to reasonable values. As RL researchers explore more complex architectures with more pieces, they introduce more hyperparameters. Tuning them often becomes a practical barrier to find signals of improvement over SOTA, perhaps limiting such research to groups with large computational budgets. Practically speaking therefore, the efficient auto-tuning of hyperparameters may be among the biggest step changes we can make to our algorithms. If the reviewers believe that self-tuning has the potential to be transformative to the practice of RL research, progress in that direction is worthy of publication.

The most significant contribution of this paper is a demonstration that self-tuning leads to big gains in performance for both STAC and STACX. We believe that achieving SOTA results in established and challenging benchmarks is **one** important way to make progress in ML research. It is easier to improve an under performing agent in a small environment than to improve a SOTA algorithm on an established benchmark. Our paper demonstrates significant empirical gains from using meta gradients in all benchmarks – the most significant gains that had been reported from using meta gradients so far. We believe that these findings would be of interest to the NeurIPS community.

We are aware that only pushing for SOTA results may lead to over fitting to certain domains and result in shallow understanding. To address these issues, we performed two types of experiments. 1. We demonstrated that the self-tuning ideas transfer from the ALE domain to control environments; we measured relative improvement from self-tuning in all the benchmarks. 2. We performed extensive ablative studies, robustness studies and visualizations of adaptivity. These experiments suggest self-tuning as many differentiable hyperparameters as possible, that self-tuning is quite robust, and that it can even discover theoretically sensible properties (contraction in V-trace).

**Auxiliary tasks and self tuning (R3 & R4)**. R4 suggested that these are orthogonal ideas, and R3 had questions regarding the ablative study. Auxiliary tasks are a good example of the above-mentioned increasing complexity of RL architectures that introduces many new hyperparameters but end up being beneficial by improving the representation power of DRL agents. This is where self-tuning comes into the picture. In Section 3 we explain that by using self-tuning, we can instead introduce auxiliary tasks while only adding a single new hyper parameter (the number of tasks). We show that *without* self-tuning, these auxiliary tasks achieve similar performance to other auxiliary tasks (246, similar to the unreal agent with 252). But, with self-tuning, there is only one hyperparameter and we achieve much higher performance. The ablative analysis in Figure 3 (b) shows what happens when we tune different subsets of the meta parameters in with auxiliary tasks (STACX, blue bars). Specifically, to answer Q2 by R3, self-tuning all the meta-parameters compared to only the discounts resulted in improving the median from 262 to 364.

**Leaky V-trace (R3 & R4).** R3 mentioned that the leaky V-trace solution seems a bit adhoc and R4 was not sure why we chose to use it over other off policy evaluation mechanism. Leaky V-trace was a contribution to making the trade-offs among bias, variance and contraction in a **differentiable** (and thus self-tunable) and interpretable form as we discuss in Section 3 and as quantified in the proof. We demonstrated that in most of the games, the Leaky V-trace parameter self-tunes from V-trace to importance sampling as training progresses and the policy changes less rapidly (the adaptivity curves). In addition, the ablative analysis confirms that trying to differentiate the thresholds directly does not perform well. More broadly, many theoretically grounded loss functions often satisfy different trade-offs and self-tuning can help to mix and adapt them. Leaky V-trace provides a simple but important evidence that its possible to do that.

**R3.** We did not address non differential hyper parameters in the paper, but a recent paper "Online Hyper-parameter Tuning in Off-policy Learning via Evolutionary Strategies" by Tang et al does.

**R4**. **Random seeds.** As a design principle we believe that given a fixed budget of compute it is more meaningful to test an algorithm across a wider range of environments than across a wider range of random seeds. The reasoning behind this is that averaging across environments has more variability than averaging across seeds, but still averages across the randomness of the algorithm. In particular in the ALE benchmark, variability across seeds is minimal, thus 3 seeds of 57 environments is a reasonable compromise that has been adapted by the community. **Prior work.** We did cite a few works on hyper parameter tuning and specifically for adapting the trace decay rate. We will do better in a revision and would be grateful if the reviewer points us to prior work they are aware of. **Evaluation in small environments.** Unfortunately, many ideas that work well in small environments do not scale to DRL benchmarks. In this work, wefocused on training a single agent across a diverse set of large environments. Self-tuning fits this setup since it allows the agent to adapt differently across environments and time.

**R6. Fig 3.** As we did for our ATARI experiments, the hyperparameters for IMPALA in Control are the same ones that were used previously in A3C. We use these values for a fair comparison. The hyperparameters that are related to STACX were kept fixed in Atari and Control. It's possible that there are slightly better hyper parameters, but we did not find the algorithm to be too sensitive to them. **Run time.** We would like to refer the reviewer to Table 4 in the supplementary where we discuss this in more detail. In short, the differences in run time are almost negligible since we are not fully utilizing our hardware. Other than that, in many cases the measure of interest is the sample complexity and not the computational complexity, thus, using more compute to gain better performance with under the same sample budget is justified. In other setups, this might not be the case.

[Meta-Review · NeurIPS 2020]

This paper shows how to automatically optimize hyper-parameters of RL algorithms (specifically IMPALA here) by gradient descent, while the agent is learning. Initial reviews were mixed, with all reviewers seeing it as a borderline paper, but trending towards rejection. However, after taking author feedback into account and discussing the pros and cons of the submission, a consensus towards acceptance emerged. Everyone (myself included) agrees that although this work is mostly incremental, it convincingly demonstrates that hyper-parameter optimization is possible on a wide range of RL tasks. This is a meaningful contribution, given how hyper-parameters of RL algorithms can be challenging (/ computationally intensive) to tweak.